# Undergraduate Student Gender, Personality and Academic Confidence

**DOI:** 10.3390/ijerph17155567

**Published:** 2020-08-01

**Authors:** Paul Sander, Jesús de la Fuente

**Affiliations:** 1Department of Psychology, Teesside University, Middlesbrough, Tees Valley TS1 3BX, UK; 2School of Education and Psychology, University of Navarra, 31009 Pamplona, Spain; jdlfuente@unav.es; 3School of Psychology, University of Almería, 04120 Almería, Spain

**Keywords:** sex differences, academic confidence, personality, university students

## Abstract

Within a socio-situational and socio-behavioural context, the relationships between the Big Five personality traits and the academic confidence of university students and how they differed by sex of the student was explored. Previous research has identified both conscientiousness and academic confidence as being linked to university performance. In respect of sex, female students have been found to score higher on all of the Big Five measures, whereas the relationship between sex and academic confidence has been mixed. Using self-report measures of personality and academic confidence from 1523 Spanish students, it was found that the female students were more confident in their grades, studying and attendance components of academic confidence and had higher scores for conscientiousness, agreeableness and neuroticism personality measures. A multiple regression analysis found that personality predicts academic confidence, with conscientiousness being the trait that statistically loaded the most strongly. This research further confirms the validity of the Academic Behavioural Confidence scale and suggests that measures of personality and, especially, academic confidence could be usefully used in student support situations to help students acquire the strategies and skills that lead to successful university study. It is suggested that further research in the area needs to include outcome or achievement measures and measures of hypothetical constructs, such as personality and academic confidence, that go beyond self-report measures.

## 1. Introduction

Whether someone is male or female is a highly salient characteristic in our social worlds, one that is used with ease to categorise people [1] and, from that, to make predictions and offer explanations for behaviour. This is well-evidenced in the history of the study of sex differences in psychology, in which the review by Maccoby and Jacklin in 1974 [2] was a seminal moment [3]. Maccoby and Jacklin found that, whilst there were a small number of reliable gender differences—for example, that boys are more aggressive and that girls have higher verbal ability—essentially, boys and girls are much more similar than they are different on the majority of psychological variables, a finding that still holds true forty-six years later [3,4,5]. Indeed, Hyde (2005) [6] proposes the Gender Similarities hypothesis, arguing from her review of 46 meta-analyses that, whilst acknowledging that there are gender differences, these can be age and context-specific. However, globally, gender differences in life’s opportunities and lived experiences are huge, in favour of men and only closing slowly [7].

### 1.1. Gender in Society

Taking a socio-situational approach, gender as a salient characteristic draws attention to the fact that, around the world, women remain seriously disadvantaged, with a gap of almost a third to closing the population weighted average gender difference [7]. The largest disparity is to be found in political empowerment, with globally less than a quarter of parliament seats occupied by women. Similarly, in the senior private sector, just over a third of managers and public sector officials are women. In the labour market, over three quarters of the men are in employment, as opposed to just over half of the women, made worse by the fact that the earnings of the women are around half that of the men in similar positions. Globally, the education gap between boys and girls has been closing, although 10% of girls between the ages of 15 and 24 remain illiterate. This is seen mainly in developing countries, but even in countries that have closed the gender education gap, it is not clear that the qualifications that the female students leave with are those that are most required for the professions that will be required in the future. Iceland, for example, despite being the most gender equal country in the world for the last 11 years, has still only closed the gender gap by 88%, meaning that it remains gender unequal [7].

The neolithic farming revolution in general, and plough farming in particular, have been linked with creating inequalities for girls and women [8,9,10]; however, that is likely to be just a part, even if a substantial part, of the explanation for the persistence of the extensive lack of opportunities for women around the world [11].

Biology has been invoked as an explanation for sex or gender differences [12,13]; by invoking the neolithic farming revolution, researchers and theorists are here talking about essentialistic, distal causes as distinct from more constructivist, proximal causes, which Wood and Early [14] synthesised into a biosocial approach to sex differences. From a socio-behavioural perspective, the salience of gender facilitates in-group/out-group formations from which readily flows in-group favouritism and out-group hostility [3,15]. These intergroup dynamics lead the debate into the realm of the influence of peer relationships and self-construal in the construction of gender differences [3,16], from which the stereotype-threat dynamic can come into play [17,18]. In a university context, these factors will combine to influence the respective approaches of male and female students to each other and to their academic studies.

A socio-linguistic perspective shows that gender is salient through being embedded in language, more extensively in some languages than others but present as an integral part of grammar in English—for instance, through gendered nouns and pronouns [19]. Thus, language draws attention to gender in situations where it is not necessary. A teacher saying “Good Morning, Boys and Girls” could just as easily have said “Good Morning, Class (or Students or Pupils)”. As Bigler [20] stresses, there is no necessity in drawing attention to the fact that some pupils or students in that the class are girls (or young women), whilst others are boys (or young men).

### 1.2. Student Gender as an Academic Variable

The salience of the sex of a student has been picked up by Sander and Sanders [21], who said that their “interest in gender differences in orientation to academic study was prompted by an accumulation of anecdotal data that male and female students seem to behave differently in relation to their academic studies” (p. 33). DiPrete and Buchmann, in the preface to their book [4], explain how their research on gender differences in education in the United States of America was triggered by their independent observations of the differential successes and the need for help of boys and girls in their children’s schools. The salience of gender has created an awareness of the differences of boys and girls, or young men and women, in educational strategies and achievements.

Socio-culturally, the disparity between boys/men and girls/women in education has reduced dramatically, second only to health [7,22]. In the United Kingdom, the attainment of female university students now outperforms that of their male counterparts [23]. A similar reversal in the education gap has been seen in the United States of America, where over half of all college students are women who are also more likely than male students to stay in education and obtain degrees and postgraduate qualifications, which can be attributed to the better social and behavioural skills in girls, which lead to a greater rate of cognitive learning and higher levels of academic investment, as measured by activities such as doing homework tasks [4,24,25]. Farsides and Woodfield (2010) and Sheard (2009) found that what students do in university, especially additional work and activities, was the best predictor of their academic outcome. This positive orientation to academic work appears to be established early in life, possibly through parental involvement [26] and shows a gender differential.

The inspection of HESA (Higher Education Statistics Agency) [27] data for 2018/19 shows that 75% of female UK students attained a first or an upper-second, as opposed to 71% of the male students. Conversely, 29% of the male students were awarded a lower-second or less against 25% of the female students, which suggests that, whilst, at times, the salience of gender is an unnecessary distraction that may lead to prejudice and discrimination through categorisation, it could also be beneficial in drawing attention to inequalities and, indeed, lost opportunities and human rights.

In Spain, recent data shows that female university students likewise outperform their male counterparts, although, following graduation, their salaries are significantly less [28]. Still, in Spain, Costa and Tabernero [29] found, in a sample of secondary school pupils, that the self-concept held by the female students exceeded that of the male students and, most importantly, that self-concept was a predictor of academic performance. An extensive report from Eurydice [30] suggested that the situation in Spain, with respect to the genders of university students, is comparable to that of the UK.

### 1.3. Gender and Academic Confidence

Martin and Phillips [31] showed that drawing attention to gender in business settings can have an adverse effect on the confidence of female employees, something that is often associated with being female [32,33]. However and in contrast, the fact that female students are now outperforming male students in UK universities arises the possibility that female students have a concomitantly greater confidence than their male counterparts, especially in courses like psychology, in which the female students considerably outnumber the male students. Sander and Sanders [34] and Sander, Putwain and de la Fuente [35] reported sex differences in university students in psychology and healthcare programmes using the Academic Behavioural Confidence (ABC) scale [34], finding that the male students were more confident in the grades that they attain, in their confidence in studying and in talking about their studies, which runs counter to the current performance of female students in university education. In-line with these findings, Sanders, Sander and Mercer [36] found that male psychology students had significantly higher self-esteem scores than their female counterparts and were significantly more confident of attaining higher grades. However, these findings showing greater confidence in male students come from relatively small samples a decade ago. Has the confidence of female students in aspects of their university degree as measured by the Academic Behavioural Confidence scale changed, especially as the female students are now more likely to know that they are outperforming male students in the UK’s higher education sector since 1992 [37] and in courses where the female students are in the majority? One of the aims of this study is to answer that question.

The gender differences in students’ performances in higher education could be coming from the way students see themselves as gendered individuals through social comparisons. The female students will know that they are the hard-working, diligent, conscientious ones who, over the years, have come to achieve better, on average, than the male students. As one male student noted: “It’s weird. It happened all through school. They just seem to have a higher drive to, the motivation to, work. Not necessarily smarter than boys, but their drive to work and keep plugging away, it’s so, it’s more in tune than boys. Boys tend to leave it ’til the last”; [36]. This difference in verbalising between the male and female students is supported by substantial literature considering the difference in verbal behaviours between male and female students, particularly in tutorial settings [38,39,40].

From years of social comparisons in school, the female students should show greater confidence in the grades that they can attain, in the studying that they do and in their likelihood to attend taught sessions. Sanders, Sander and Mercer [36] go on to say: “I think they (female students) have a bigger need for order, you know ‘cos today we had, uh, sort of a lesson today, and we, we had to write ideas down, and in our group, we gave all the writing and the organising to put it down on paper to the girls. ‘Cos we knew, to be honest, they’d be neater. They’d plan it out better (agreement from the other group members), but all the ideas and things to write down came from us, but they could present it really well, but I think we were sort of turning up all the ideas” (pp. 9–10). From this, it could be predicted that the male students will show greater verbalising confidence than the female students.

### 1.4. Gender and Personality

Is the diligent, conscientious, committed nature of female students something that is part of them, part of their personality? Research has shown that the Big Five personality factors: openness, conscientiousness, extraversion and agreeableness positively correlate with university academic performance and neuroticism negatively or poorly [41,42,43]. Conscientiousness has been found to be the strongest predictor, albeit one moderated by the subject of study [44] and the way in which personality is measured [45]. Farsides and Woodfield [24] found that the personality trait of openness, along with learning as a motive for being at university and in doing formative exercises and activities, to be the biggest predictors of student achievement, the latter supported by Sheard (2009). Gender differences were found in openness, agreeableness and neuroticism, but Farsides and Woodfield [24] concluded that what students do whilst at university, activities such as doing formative work that does not contribute to their overall degree performance, is more important than individual differences between students upon arrival in determining their eventual undergraduate academic success. This leads to the prediction that the Big Five personality trait of conscientiousness should be more predominant in the female students than in the male students, as this could be a driver of the commitment [25], motives and learning activities [24] that underpin higher achievement in university studies. A student’s gender may be associated with those and other attributes of successful university study, but gender, per se, is not causal.

### 1.5. Aims and Hypotheses

From the theory and research presented, it can be predicted that the Big Five traits of openness, conscientiousness, extraversion, and agreeableness should be positively correlated with students’ academic confidence as measured through the Academic Behavioural Confidence scale, and neuroticism should be negatively correlated. The female students should have higher scores in all measures. Finally, it can be predicted that personality should have a causal link to academic confidence, being a more general trait.

In summary, the objective of this research is to test the three hypotheses as presented below, which operationalise an analysis of the relationships between the Big Five personally traits and the four subscale measures of the academic confidence of university students using dates from two validated psychometric measures. At the same time, the research seeks to identify any gender differences that might arise in personality and academic confidence. From this, any influence of gender and personality of undergraduate students will be established.

There will be a difference in the Academic Behavioural Confidence subscales of grades, verbalising, studying and attendance and in the Big Five traits of openness, conscientiousness, extraversion, agreeableness and neuroticism between male and female students.The Big Five traits of openness, conscientiousness, extraversion, and agreeableness will positively correlate with the Academic Behavioural Confidence subscales. The Big Five trait of neuroticism will negatively correlate with the Academic Behavioural Confidence.The Academic Behavioural Confidence will be predicted from sex, conscientiousness, openness, neuroticism, extraversion and agreeableness.

## 2. Method

### 2.1. Participants

The sample comprised undergraduate students enrolled in Psychology, Primary Education or Educational Psychology degree programmes from two universities in Spain. The data used here comes from two different datasets, the first with data collected between 2012 and 2014 and the second from 2015 to 2018. Both of these datasets had in common the variables sex, age and academic confidence, but only the more recent datasets included personality measures. The data collection process was the same in both datasets, as were the degree courses and the universities. The participants’ ages in years and sex are shown in Table 1, along with the relevant variables that were measured and, thus, had values in the datasets. In completing the psychometric scales, students were asked, amongst other demographic questions, whether they were male or female, without differentiating between the very different constructs of sex and gender. As the UK Office for National Statistics [46] says, these terms are often used interchangeably, and that is the sense in which they are used here. There was no attempt to differentiate at the data-gathering stage, and the variable was labelled sex, but much of the literature talks about gender. In the context of this article, the terms are interchangeable unless otherwise stated.

Differences between male and female participants for academic confidence were calculated from an aggregate dataset (*n* = 1838) in which 48 participants failed to record their sex, whereas sex differences for personality come solely from the second, more recent dataset, as do the correlations between personality and academic confidence. Whilst acknowledging that it would be preferable to have roughly equal numbers of male and female students for a consideration of sex differences in personality and academic confidence, the proportion of females to males and the age profiles of the participants match the profile of student cohorts on the degree into which they were enrolled and is therefore representative of students in such programmes.

### 2.2. Instruments

Big Five Questionnaire [47] used to measure personality was based on a version by Barbaranelli et al. [48], which was adapted and revalidated for young university students [49]. The scale contains 67 statements, and a previous Confirmatory Analysis (CFA) reproduced a penta-factorial structure corresponding to the Model of the Big Five [50]. The results have shown adequate psychometric properties and acceptable adjustment rates. The confirmatory model second order showed a good fit (chi-square = 38.273; degrees of freedom (20–15) = 5; *p* < 0.001; Normed Fit Index, NFI = 0.939; Relative Fix Index, RFI = 0.917; Incremental Fix Index, IFI = 0.947; Tucker-Lewis Index TLI = 0.937, Comparative Fit Index, CFI = 0.946; Root Mean Square Error of Approximation, RMSEA = 0.065 and HOELTER index = 2453 (*p* < 0.05) and 617 (*p* < 0.01)), and the internal consistency of the total Scale is good (alpha = 0.956; Part 1 = 0.932 and Part 2 = 0.832; Spearman-Brown = 0.962 and Guttman = 0.932).

Academic confidence was measured by the 24-item Academic Behavioural Confidence (ABC) Scale in a Spanish-validated version [51]. The ABC scale is a psychometric measure of the confidence of undergraduate students in their anticipated study-related behaviours in a largely lecture-based course. The scale items in English and Spanish can be found as the appendix to Sander et al. [51]. Previous work has shown a four-factor model (confidence in attaining grades, studying, attending classes and discussing course material) with adequate reliability and validity [34,51,52]. The scale requires students to respond to a question stem (“How confident are you that you will be able to... “) on a five-point scale (1 = “not at all confident” to 5 = “very confident”) for items such as “...manage your workload to meet coursework deadlines” and “...write in an appropriate academic style”. A higher score therefore indicates greater confidence in self-efficacious study skills or behaviours. Table 2 maps the scale items onto the subscales of the Academic Behavioural Confidence scale and records the internal reliability of each.

The internal reliability measures for the subscales grades and verbalising are good and that for studying is acceptable. The alpha value of 0.625 for a reduced attendance scale seems poor, but as Field [53] points out, the size of the alpha is dependent also on the number of items in the scale. The more items there are, the higher the alpha is likely to be, so with just two items, one could expect a low alpha value. The absolute correlation between items 6 and 18 is 0.458. Thus, the attendance scale is seen as acceptable. Item 24 was a very specific item, attend tutorials, which was largely not applicable to those Spanish students. The McDonald’s **ω** (omega) [54] coefficients match those of Cronbach’s alpha.

A confirmatory factor analysis was performed to validate the four subscale structures of the Academic Behavioural Confidence scale, as shown in Table 2, but with item 24 deleted, as, out of the 1838 participants, only 424 gave an estimate of their confidence. Item 24 in the ABC scale is unique in that it asks specifically for confidence in attending tutorials, and increasingly, courses do not offer traditional tutorials but, rather, seminar or workshop sessions or even drop-in sessions. The renaming items in this scale, 6—attend most taught sessions and 18—be on time for lectures, would seem both statistically and with face validity to cover this possibility. The resultant fit statistics (see Table 3) were supportive of the four subscale structures.

### 2.3. Procedure

Participants voluntarily completed the scales in Spanish using an online platform [55] (http://www.estres.investigacion-psicopedagogica.com/english/seccion.php?idseccion=1). All students gave their informed consent through an online signature that is required when creating an account on the platform before any questionnaires are completed. A range of specific teaching-learning processes was evaluated, covering different university subjects over a two-year period, and these included the scales in question here. To avoid fatigue, students were invited to complete only one questionnaire at two different times of each week during a semester. For their participation, they were provided with a Certificate of Participation in Research as an incentive to maintain motivation and recognise their efforts. In the Spanish-speaking world, such certificates can be very beneficial to students alongside their CV. The procedure was approved by the respective ethics committees of the two universities in the context of two R&D Projects.

### 2.4. Data Analysis

Pearson correlation bivariate correlations explored the relationship between personality and academic confidence. Student *t*-tests were used to test for differences between the male and female students for the five personality traits and the four academic confidence subscales. Finally, multiple regression was used to model the effects of personality on academic confidence. All of these analyses were conducted in SPSS (v.26) [56].

Confirmatory factor analysis was used to provide evidence of the factorial validity of the Academic Behavioural Confidence scale. Model fit was assessed in Onyx [57] by examining the chi-square to a degrees of freedom ratio, the Tucker-Lewis Index (TLI) and Comparative Fit Index (CFI), which, for a good model fit, should be greater than 0.90, and the RMSEA statistic, with a value less than 0.06 [58,59]. Scale reliability was assessed using Cronbach’s alpha in SPSS (v.26) and McDonald’s **ω** (omega) [54] in JASP [60].

## 3. Results

### 3.1. Sex Differences in Academic Confidence and Personality

This first section of the results addresses hypothesis 1. Using the combined dataset reduced to include just those students who had completed the academic confidence scale, the differences in scores between the male students and the female students were assessed. The means presented in Table 4 show the female students as more confident in their grades, in studying and in attending, but less confident than the male students in verbalising. All of these differences are statistically significant, with small effect sizes, with the exception of the effect size associated with the difference between male and female students in verbalising, which is a medium effect size.

Table 5 reports the differences between male and female students in each of the five personality measures from the data from the smaller, more recent dataset, as the only one that collected the personality measures. Table 5 shows that there are statistically significant differences between the male and female students for the traits conscientiousness, neuroticism and agreeableness, all with a small effect size. Specifically, the female students show higher scores for all three traits.

In both Table 4 and Table 5, the effect size shown is Hedges’ gas; it corrects for different sample sizes, as presented here. In each Table 4 and Table 5, multiple comparisons are being made, meaning that a Bonferroni correction is needed. With an adjusted alpha of 0.01, all differences shown remain significant. Taking this adjusted alpha value of 0.01 and applying the Benjamini–Hochberg (BH) procedure, these differences remain significant [61,62]. The BH-adjusted *p*-values are shown in Table 4 and Table 5.

### 3.2. Correlations between Academic Confidence Subscales and the Big Five Personality Traits

Table 6 presents the correlations between the students’ personality and academic confidence, regardless of sex. It was proposed in the second hypothesis that the Big Five traits of openness, conscientiousness, extraversion and agreeableness would positively correlate with the Academic Behavioural Confidence subscales. The Big Five trait of neuroticism will negatively correlate with the Academic Behavioural Confidence.

It can be seen in Table 6 that there are statistically significant correlations between each of the five personality measures and the four academic confidence measures and that these are positive, with the exception of the correlations between neuroticism and all the academic confidence measures, which are all negative. Notable in these results are the magnitudes of some of the correlation coefficients. Specifically, (i) the correlations between extraversion and verbalising (0.342); (ii) conscientiousness and grades (0.525), studying (0.623) and attendance (0.411) and, finally, (iii) openness with grades (0.338), studying (0.33) and verbalising (0.353).

### 3.3. Level of Academic Confidence from Sex and Personality

In respect of the third hypothesis, a multiple regression analysis was performed to explore the impact of sex of the student and their personality, as measured by conscientiousness, openness, neuroticism, extraversion and agreeableness on each of the four subscales of the Academic Behavioural Confidence scale. The results of these linear regression analyses are shown in Table 7, where it can be seen that each of the four models, one for each of the academic confidence subscales, is significant. The adjusted R square values (R^2^) suggest that the academic confidence subscale studying is most affected by the sex of the student and their personality profile. Of the five personality traits, conscientiousness and openness have the greatest significant impacts across all measures of academic confidence, with the magnitude of the standardised beta coefficients for the personality trait conscientiousness being notable as is the consistent negative loading of the trait neuroticism. In none of the four models is sex itself not a significant contributor to any of the academic behavioural confidence subscales, but its inclusion moderated the effects of the individual personality variables. This was seen by the sequential inclusion of the personality variables in a regression model that started with just gender. The personality variables were entered in order of the increasing correlation coefficient with the relevant Academic Behavioural Confidence (ABC) subscale. For example, for the ABC subscales grades and studying as dependent variables, sex remained a significant weight until the inclusion of the personality trait conscientiousness.

In summary, the results have shown that personality and academic confidences were correlated as predicted, and that the female students were more confident in each of the academic confidence measures, with the exception of verbalising. The female students also had higher levels of the personality measures of conscientiousness, agreeableness and neuroticism. Finally, the regression analyses show that each of the subscales of Academic Behavioural Confidence may be predicted from the sex and personality profile of the student, with the greatest impact being the personality trait of conscientiousness on the academic confidence measure of studying.

## 4. Discussion

Three hypotheses were put forward, which will be examined in turn. For the first, there will be a sex difference in the Academic Behavioural Confidence, and in the Big Five personality measures, considerable differences were found. There were significant differences in the ABC scores for each of the four subscales, with the female students showing higher levels of confidence in their grades, studying and attendance, as predicted. Additionally, as predicted, the male students were more confident than the female students in their verbalising skills. This difference had a medium effect size, whilst the others were all small, but that is still one in which there is a real effect [6]. Although these findings conflict with previous sex difference results using the Academic Behavioural Confidence scale [34,36], the research presented here has used a much larger sample size. With this substantial methodological improvement, it is suggested that note has to be taken of the findings in gender differences presented here, which also align with the research findings presented in the introduction.

For the Big Five measures, there were significant differences between male and female students’ trait scores for conscientiousness, neuroticism and agreeableness, with medium effect sizes, which are in-line with Mac Giolla and Kajonius [6] and Schmitt et al. [12]. The traits of extraversion and openness showed no significant difference, whereas Mac Giolla and Kajonius [60] also found that women scored higher than men for the traits of extraversion and openness, and Schmitt et al. [12] reported higher levels of openness. Both of these studies [12,63] were exploring the relationship between gender differences in personalities around the world and the gender equality index of the country, finding that, as gender equality increases, so does gender differences in the Big Five measures. With the data reported in this study coming from Spanish undergraduate students and Spain ranking eighth in the Global Gender Gap Index 2020 rankings [7], gender differences in personality are to be expected and were found.

The second hypothesis predicted a positive correlation between the Big Five traits of openness, conscientiousness, extraversion and agreeableness and the Academic Behavioural Confidence subscales, with the correlations being positive with the exception of neuroticism, which would negatively correlate with the Academic Behavioural Confidence. The findings were that extraversion positively correlated with the ABC subscales, apart from attendance. Conscientiousness positively correlated with all four ABC subscales and with correlation coefficients that are surprisingly high for research in the social sciences [64], with the exception of the relationship with verbalising. Neuroticism correlated negatively but lowly with all the ABC subscales, as predicted, and agreeableness and openness correlated significantly and positively. Given the relatively large sample size of 527 for those calculations, small but significant correlations are not a surprise, but the magnitude of the positive correlation between conscientiousness and the ABC subscales of grades, studying and attendance are noteworthy and inline with previous findings on the importance of high conscientiousness scores and academic achievement in universityies [41,42,43,44]. It could also be posited that the doing of formative exercises that Farsides and Woodfield (2010) [24] and Sheard [25] found to predict academic success are the hallmark of the conscientious student, as is going to university with the intention to learn [24]. Thus, the findings reported here are in-line with those from previous research and help, if only in a small way, to build a profile of the typical, successful university student. Finally, and driven by the third hypothesis, personality was found to be a predictor of overall academic confidence, with conscientiousness being one of the three Big Five traits that statistically added to that prediction. Conscientiousness had, by far, the biggest impact on academic confidence, as shown through the correlations and the regression analysis. The differences in the academic confidence and personality measures between the male and female students were small but as predicted from previous research. Whilst gender does not significantly weigh on any of the four Academic Behavioural Confidence subscales in the regression analyses, its impact being overshadowed by the impact of the personality variables, it is, nonetheless, a variable that has to be considered in the educational processes.

In specific relation to students, teachers and day-to-day education, the results make some suggestions for guiding the dynamics of the teaching and learning processes. The data shows that female students are more confident of obtaining higher grades, of doing the necessary studying to achieve them and of attending classes, but they are less confident than their male peers in talking about their studies. In respect of their personalities, the female students are more conscientious, more agreeable but have higher neuroticism scores. Thus, the female students are not uniformly better in their orientation to either studies than the male students but, across the board, have more qualities and strategies than the male students that will lead to higher levels of academic achievement. This raises a number of questions: Firstly, should the female students be encouraged to talk more about their subject of study with academic staff in tutorials, seminars, lectures and other conversations, and should they be encouraged to worry less, to embrace fewer of the symptoms of neuroticism that they outscore the male students on? Secondly, should time, effort and attention be given to the male students to support them to orient towards their academic studies in a way comparable to their female counterparts? Thirdly, the results show that, whilst gender on its own does have an effect on both personality and academic confidence, when put into regression models, the effect of gender is removed, sooner or later, by personality variables—in particular, conscientiousness and openness. However, personality itself is not malleable, certainly over the short term, within an educational establishment, even if it does support the findings of Gorard [26] that the home environment and, in particular, the support that it offers is the biggest single predictor of school-level educational achievement. Thus, whilst an overall political goal might be to stimulate parents and other significant adults to support, encourage and guide young people’s education, regardless of their gender, it is not a level of analysis that is going to help the educator with a group of students. At that level, the answer to the first of the questions posed above might be that, yes, female students would be encouraged, guided and supported in talking more about the study of their subject and to worry less about it. Similarly, the answer to the second question could be yes, male students should be explicitly encouraged, guided and supported to be more like their female counterparts in the approach they take to their studies, doing what is necessary to be confident that they will get good grades, confident in their study behaviours and in the attendance of scheduled sessions, to which might be added and, in also, doing, other, additional formatives. Thus, the results offer clear guidance for teachers and those who support students in general in universities, and maybe elsewhere, in how best to work with a student’s gender in support of academic achievements.

Taken together, the female students were more confident in their grades, studying and attendance, arguably the profile of a good student from a teacher’s perspective and a profile that could lead the student to do those extra, additional, set activities [24,25]. The female students had higher scores for conscientiousness, agreeableness and neuroticism, and, as already noted, conscientiousness is known to be a reliable predictor of academic success [41,44].

The findings presented here are important, not just because they substantiate a range of previous research in the area of sex differences in personality and confidence in educational settings but because they help to build a profile of achievement in higher education, which can be used by personal tutors and other academic support systems in higher education to guide the development of key skills in students. They support a socio-situational and a socio-behavioural approach to gender differences in education, arguing that both—distally and proximally, the environment and, especially, the social environment—create the conditions for female students to come to see themselves as successful and, through social categorisation, see themselves as having better, more successful and thorough strategies and behaviours for achieving in higher education. To fully understand the processes involved in students’ emergences as successful learners, one has to look behind the proxy measures of personality and academic confidence to the socio-situational factors that set young people on a pathway to success in education and why, perhaps through stereotype threats [18], that is more likely to happen for girls and young women in education than for boys and young men. This is a very important area to explore further. The findings presented here are also important, as they offer a way—just a very small way, admittedly—to address the gender gap that exists across the world. Our female students should be celebrated for their successes and used as role models.

### Limitations

The Big Five trait of conscientiousness as a hypothetical construct cannot be considered truly causal, per se, in determining the successful study behaviours of students. The same applies to academic confidence. These traits are more likely to be proxies for something situational that created the behavioural characteristics of conscientiousness, any genetic contribution to personality aside. One possible candidate is parental involvement in children’s education, which Gorad, See and Davies [26] found from an extensive review to be the strongest of a wide range of measures that predicted academic school performance. The extent to which parental involvement remains a prime influencer of academic attainment at university is an avenue for future research, but, to speculate, one can see how the appropriate support and encouragement of children by their parents during their school years could lay the foundation and pave the way for a commitment to education that could be encapsulated in the hypothetical constructs of conscientiousness and academic confidence. In short, the focus needs to be on the social environments constructed for boys and girls as they grow up, with particular attention paid to culture and social stereotypes [5].

Other areas prime for future research in this area include measures of actual academic achievements. The inclusion of other reported measures, in addition to self-report measures of variables such as personality and academic confidence [44], should also be considered. That all of these are absent in the research considered here is a drawback to this research. Finally, it would also be wise to follow Weisberg, Deyoung and Hirsh [65] and measure the Big Five traits at the aspect level in order to follow more carefully the gender differences in personality characteristics that may be lost at the macro level of the five traits.

## 5. Conclusions

The data presented here shows the strong relationship between students’ personality and their academic behavioral confidence with the Big Five trait of Conscientiousness being particularly strong but not exclusively related to academic confidence. Whilst gender did not appear as a significant contributor to academic confidence in the regression analyses, clear gender differences were found in academic confidence with the female students showing higher confidence that the male students for Grades, Studying, and Attendance. The male students had greater confidence in verbalizing. There were also significant gender differences for personality with the female students showing higher scores for conscientiousness, neuroticism, and agreeableness. It was argued that the strength of the contribution of conscientiousness in the regression models attenuated the effect of gender.

## Figures and Tables

**Table 1 ijerph-17-05567-t001:** Participant profile for the two component datasets.

Dataset	Sex		Age (Years)				Variable Measures in the Datasets
	Male	Female	Mean	Min	Max	SD	
2012–2014	386	1020	23.05	18	51	4.49	Sex, Academic Confidence
2015–2018	83	301	21.61	17	58	5.42	Sex, Academic Confidence, Personality
Total	469	1321					
Total students with known sex	1790						

**Table 2 ijerph-17-05567-t002:** Internal reliability of the 4 factors of the Academic Behavioural Confidence scale.

Scale	Scale Items	Scale Internal Reliability	
		Cronbach’s Alpha	McDonald’s ω (omega)
Grades	2, 7, 15, 16, 20, 23	0.815	0.815
Studying	1, 4, 21, 22	0.703	0.710
Verbalising	3, 5, 8, 10	0.827	0.834
Attendance	6, 18, 24	0.197	0.289
	6, 18	0.625	0.623

**Table 3 ijerph-17-05567-t003:** The fit statistics for a confirmatory factor analysis of the 4 factor structurea of the Academic Behavioural Confidence scale.

Chi-Square	759.414
DF	98
RMSEA	0.061
CFI	0.935
TLI	0.921
SRMR	0.045

DF: degrees of freedom; RMSEA: root mean square error of approximation; CFI: comparative fit index; TLI: Tucker-Lewis Index; SRMR: standardised root mean squared residual.

**Table 4 ijerph-17-05567-t004:** Means and standard deviations for each of the academic confidence sub-scales by sex of student.

Gender Statistic	Academic Behavioural Confidence			
	Grades	Studying	Verbalising	Attendance
Male Mean *n* = 469 SD	3.93 0.65	3.73 0.71	3.32 0.90	4.17 0.88
Female Mean *n* = 1321 SD	4.02 0.63	3.91 0.69	3.03 0.95	4.34 0.83
*t*-test, df, *p*	2.65, 1788, 0.008 *	4.90, 1788, 0.0005 *	5.79, 1788, 0.0005 *	3.73, 1788, 0.0005 *
BH-Adjusted *p*	0.0175	0.0075	0.0075	0.0075
Effect size, Hedges’ gas	0.14	0.26	0.31	0.20

* Difference is significant at the 0.01 level (2-tailed) with a Bonferroni correction for multiple comparisons. BH: Benjamini–Hochberg and df: degrees of freedom.

**Table 5 ijerph-17-05567-t005:** Means and standard deviations for each of the academic confidence subscales by sex of the student.

Gender Statistic	Extraversion	Conscientiousness	Neuroticism	Agreeableness	Openness
Male Mean *n* = 113 SD	3.66 0.57	3.50 0.59	2.42 0.66	3.81 0.55	3.51 0.55
Female Mean *n* = 414 SD	3.61 0.57	3.74 0.59	2.64 0.66	4.02 0.48	3.51 0.48
*t*-test, df, *p*	ns	3.65, 498, 0.0005 *	3.13, 503, 0.002 *	3.85, 494, 0.0005 *	ns
BH-Adjusted *p*		0.0075	0.015	0.0075	
Effect size, Hedges’ gas		0.41	0.33	0.42	

* Difference is significant at the 0.01 level (2-tailed) with a Bonferroni correction for multiple comparisons. ns—non-significant.

**Table 6 ijerph-17-05567-t006:** Correlation coefficients for the personality factors and the academic confidence subscales.

	Academic Confidence			
Personality Trait	Grades	Studying	Verbalising	Attendance
Extraversion	0.215 **	0.226 **	0.342 **	0.003
Conscientiousness	0.525 **	0.623 **	0.221 **	0.411 **
Neuroticism	−0.172 **	−0.142 **	−0.175 **	−0.117 **
Agreeableness	0.278 **	0.307 **	0.171 **	0.238 **
Openness	0.338 **	0.33 **	0.353 **	0.102 *

* Correlation is significant at the 0.05 level (2-tailed) ** Correlation is significant at the 0.01 level (2-tailed).

**Table 7 ijerph-17-05567-t007:** Analysis of sex and personality on academic confidence by subscales.

	Academic Behavioural Confidence			
Statistic	Grades	Studying	Verbalising	Attendance
F-Value, df: (6, 426)	28.843 **	42.963 **	16.971 **	18.084 **
R^2^	0.279	0.368	0.182	0.192
Standardised Beta Coefficients				
Personality Trait				
Openness	0.106 *	0.041	0.256 *	−0.132 *
Conscientiousness	0.452 *	0.587 *	0.067	0.464 *
Extraversion	−0.058	−0.040	0.209 *	−0.190 *
Agreeableness	0.003	0.001	−0.086	0.139 *
Neuroticism	−0.110	−0.046	−0.113	−0.001
Sex	0.076	0.051	−0.088	−0.023

* Significant at *p* < 0.05. ** Significant at *p* < 0.0005. R^2^: adjusted R square values.

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
