# Peer review of "Undergraduate Student Gender, Personality and Academic Confidence"

_ijerph, 2020, doi:10.3390/ijerph17155567_

Round 1

Reviewer 1 Report

Here are some suggestions: #1 Please clarify what does "Mean" means in Table1. Mean of what? What does "4.49 Sex, Academic Confidence" and ”5.42 Sex, Academic Confidence" under the heading variables mean? #2 2.2 Instruments: how many items in the instrument BFQ? What data you included for CFA, both data sets? Same questions for the second instrument Academic confidence. #3. You might want to explain what are grades, studying, verbalizing, and attendance, the four headings in table 2. #4 Line 253, CFI 0.935= , should it be CFI=0.935? #5 Line 255, SMRM=7,749, It does not seem right. Should SMRM less than 1? Ideal SMRM should be less than 0.08 (Browne and Cudeck 1993) #6 You might want to explain why you choose Hedge's g as effect size other than Cohen's d. Normally g is choosen due to small sample size, e.g., less than 20. But you have over 100 in each good. #7 In Table 5, if grades, studying, verbalsign, and attendance are four factors under academic confidence, you are running multiple comarison -- BH correction is need. ---Same suggestions for Table 6. #8 in Table 7, if you prefer to use * and** to represent significant level, please do the same thing for Table 5 and Table 6. #9 Not very sure why multiple regression was chosen for examining the correlation among grades studying, verbalsing, attendance, extraversion...., CFA might examine all of them at once ---- just one consideration.

Author Response

Reviewer 1 Here are some suggestions: 

#1 Please clarify what does "Mean" means in Table1. Mean of what? What does "4.49 Sex, Academic Confidence" and ”5.42 Sex, Academic Confidence" under the heading variables mean? 

Thank you for drawing attention to this.  I have changed table 1 to make clear that age is measured in years and that the final column is listing the relevant variables that existed in each of the two data sets as they were no identical as only the second data set had personality measures.  I have also amended the last sentence of the paragraph before the table to make this clearer, highlighted.

#2 2.2 Instruments: how many items in the instrument BFQ? 

67 This has been added in page 5 - highlighted

What data you included for CFA, both data sets? 

These findings came from earlier research which I have now referenced which I should have done in the first place (highlighted on page 5).  Apologies.  

Same questions for the second instrument Academic confidence. 

24 items - highlighted on page 5.  The data used for the confirmatory factor analysis of the ABC scale was both data sets.  However the second reviewer was suggesting that this be correctly removed as it was not an aim of the research.  I will address this further in response to the second reviewer’s comments but suffice it to say here that the CFA of the ABC scale came from Sander et al (2011)

#3. You might want to explain what are grades, studying, verbalizing, and attendance, the four headings in table 2. 

I have amended the title of the table and made clear that the 4 sub-scales belong to the ABC scale in the table itself.

#4 Line 253, CFI 0.935= , should it be CFI=0.935? 

Yes it should and we have to apologise profoundly for that. 

#5 Line 255, SMRM=7,749, It does not seem right. Should SMRM less than 1? Ideal SMRM should be less than 0.08 (Browne and Cudeck 1993) 

yes, absolutely correct and to be honest, I am not quite sure how that statistic got in there.  However, I have written the statistic name correctly and given it it’s correct value of .045

#6 You might want to explain why you choose Hedge's g as effect size other than Cohen's d. Normally g is choosen due to small sample size, e.g., less than 20. But you have over 100 in each good. 

On page 7 a sentence has been added (highlighted) which makes clear why hedges’ g was chosen:  In both tables 5 and 6, the effect size shown is Hedges’ g as it corrects for different sample sizes as present here.

#7 In Table 5, if grades, studying, verbalsign, and attendance are four factors under academic confidence, you are running multiple comarison -- BH correction is need. ---Same suggestions for Table 6. 

Assuming that this comment refers to a Bonferroni correction for multiple comparisons, a sentence has been added on page 7: n each of tables 5 and 6 multiple comparisons are being made meaning that a Bonferroni correction is needed.  With an adjusted alpha of .01 all differences shown remain significant.

#8 in Table 7, if you prefer to use * and** to represent significant level, please do the same thing for Table 5 and Table 6. 

“*Difference is significant at the 0.01 level (2-tailed) with a Bonferroni correction for multiple comparisons” has been added to the foot of each table to maintain style.  These additions have been highlighted.

#9 Not very sure why multiple regression was chosen for examining the correlation among grades studying, verbalsing, attendance, extraversion...., CFA might examine all of them at once ---- just one consideration. 

Thank you.  This issue was also raised by the second reviewer and will be addressed there.  Suffice it to say that completely new analyses are presented here with multiple regression by each of the sub-scales of the academic behavioural confidence scale

Reviewer 2 Report

All the comments have been included in the attached document

Author Response

Please see the PDF file that I have attached to explain how I have engaged with each and every one of the thoughtful and constructive points raised.  I am very grateful to you for the considerable effort you clearly put into reviewing this piece of work.

Round 2

Reviewer 1 Report

BH correction refers to Benjamini-Hochberg Procedure. You might want to refer to

https://www.statisticshowto.com/benjamini-hochberg-procedure/

So you might need to have a p and a BH adjusted p to report.

BH correction is different from Bonferroni correction.

Author Response

Review 1

BH correction refers to Benjamini-Hochberg Procedure. You might want to refer to

https://www.statisticshowto.com/benjamini-hochberg-procedure/

So you might need to have a p and a BH adjusted p to report.

BH correction is different from Bonferroni correction.

I should have checked for which I apologise.  I have now computed the BH adjusted p-values and reported them in tables 4 and 5 along with adding a sentence or two to the end of the paragraph above table 4.  This additions have been highlighted in yellow.

Reviewer 2 Report

The authors have considered virtually all of the suggested changes, with a few exceptions:
- Although it is clear in the response to the review that the authors opt for the first suggested option regarding the objective of the article, the objective is still not made explicit at the end of the theoretical section (or next to the hypotheses).
- I do not find that anything has been changed in the text regarding the following suggestion: "In any case, such perception of women is not better in all dimensions. This data serves to confirm what the study by Sander et al. (2009) seems to point out, but what practical implications does it have? Should the dimension in which women score the worst be reinforced or, as it does not have as much weight on "academic confidence", is it not a priority? From an educational perspective, both family and school, the personality traits are not so modifiable, therefore, I understand that the interesting thing would be to work on the conformation of gender as well as the variables of behaviour, as the dimensions of the "academic confidence".
Furthermore, having included new information, I have a new doubt about the results:
- I find the interpretation of the multiple regressions somewhat dubious, since the results are not statistically significant. For example, sex does not have a significant weight on any of the dimensions of the ABC. This issue should also be reflected in the discussion, since it is clear that personality traits have an important weight in academic behavior, but this is not the case with sex; there would only be an indirect relationship between sex and academic behavior, which could be tested in future work using regression models (partial or complete).

Author Response

Review 2

The authors have considered virtually all of the suggested changes, with a few exceptions:

  • Although it is clear in the response to the review that the authors opt for the first suggested option regarding the objective of the article, the objective is still not made explicit at the end of the theoretical section (or next to the hypotheses). 

I have added a short paragraph at the end of the introduction which I hope makes clear the objectives of the study as subsequently spelt out in the hypotheses.  Hopefully this paragraph serves the identified need of a link between the theory and research presented in the introduction and the hypotheses themselves.

  • I do not find that anything has been changed in the text regarding the following suggestion: "In any case, such perception of women is not better in all dimensions. This data serves to confirm what the study by Sander et al. (2009) seems to point out, but what practical implications does it have? Should the dimension in which women score the worst be reinforced or, as it does not have as much weight on "academic confidence", is it not a priority? From an educational perspective, both family and school, the personality traits are not so modifiable, therefore, I understand that the interesting thing would be to work on the conformation of gender as well as the variables of behaviour, as the dimensions of the "academic confidence”.

The end of the third paragraph of the discussion has been extended and a new paragraph added below to further the discussion raised here.  That discussion includes a brief consideration of the non-significant beta coefficient of sex in the regression models.

Furthermore, having included new information, I have a new doubt about the results:

  • I find the interpretation of the multiple regressions somewhat dubious, since the results are not statistically significant. For example, sex does not have a significant weight on any of the dimensions of the ABC. This issue should also be reflected in the discussion, since it is clear that personality traits have an important weight in academic behavior, but this is not the case with sex; there would only be an indirect relationship between sex and academic behavior, which could be tested in future work using regression models (partial or complete).

Furthering the comments I made above in relation to sex as a variable in the regression models, exploring the impact of Sex and each of the 5 personality variables on each of the 4 academic behavioural confidence variables one by one was illuminating but not sufficiently significant to occupy a considerable amount of print space in a journal article.  For each of the four ABC sub-scales, when sex was entered alone into a regression model it was, as one would expect, as it is effectively a t-test, significant.  Sequentially adding the personality variables in order of the least important as measured by the correlation coefficient to the most important, one could watch the effect of gender being removed.  Thus with reference to table 6, gender remained a significant variable until Conscientiousness was added for both of the ABC sub-scales, Grades and Studying.  Its contribution to Verbalising and Attendance was less but could still be seen.

I have attempted to clarify this in the added text in the paragraph in section 3.3 of the results, just before table 7.  I hope that in doing this I have addressed your concern without providing huge tables showing the progressing models for each of the ABC sub-scales.